# A 3 Year Longitudinal Prospective Review Examining the Dietary Profile and Contribution Made by Special Low Protein Foods to Energy and Macronutrient Intake in Children with Phenylketonuria

**DOI:** 10.3390/nu12103153

**Published:** 2020-10-15

**Authors:** Anne Daly, Sharon Evans, Alex Pinto, Catherine Ashmore, Júlio César Rocha, Anita MacDonald

**Affiliations:** 1Dietetic Department, Birmingham Children’s Hospital, Steelhouse Lane, Birmingham B4 6NH, UK; evanss21@me.com (S.E.); alex.pinto@nhs.net (A.P.); catherine.ashmore@nhs.net (C.A.); anita.macdonald@nhs.net (A.M.); 2Nutrition and Metabolism, NOVA Medical School, Faculdade de Ciências Médicas, Universidade Nova de Lisboa, 1169-056 Lisboa, Portugal; rochajc@nms.unl.pt; 3Centre for Health Technology and Services Research (CINTESIS), 4200-450 Porto, Portugal

**Keywords:** phenylketonuria, PKU, glycomacropeptide, special low protein foods, macronutrient intake, protein substitute

## Abstract

The nutritional composition of special low protein foods (SLPFs) is controlled under EU legislation for ‘Foods for Special Medical Purposes (FSMP)’. They are designed to meet the energy needs of patients unable to eat a normal protein containing diet. In phenylketonuria (PKU), the macronutrient contribution of SLPFs has been inadequately examined. Aim: A 3-year longitudinal prospective study investigating the contribution of SLPFs to the macronutrient intake of children with early treated PKU. Methods: 48 children (27 boys) with a mean recruitment age of 9.3 y were studied. Semi-quantitative dietary assessments and food frequency questionnaires (FFQ) were collected three to four times/year for 3 years. Results: The mean energy intake provided by SLPFs was 33% (SD ± 8), and this figure was 42% (SD ± 13) for normal food and 21% (SD ± 5) for protein substitutes (PS). SLPFs supplied a mean intake of 40% carbohydrate (SD ± 10), 51% starch (SD ± 18), 21% sugar (SD ± 8), and 38% fat (SD ± 13). Fibre intake met 83% of the Scientific Advisory Committee on Nutrition (SACN) reference value, with 50% coming from SLPFs with added gums and hydrocolloids. Low protein bread, pasta and milk provided the highest energy contribution, and the intake of sweet SLPFs (e.g., biscuits, cakes, and chocolate) was minimal. Children averaged three portions fruit/vegetable daily, and children aged ≥ 12 y had irregular meal patterns. Conclusion: SLPFs provide essential energy in phenylalanine restricted diets. Optimising the nutritional quality of SLPFs deserves more attention.

## 1. Introduction

In phenylketonuria (PKU), deficiency or reduced activity of the phenylalanine hydroxylase enzyme (PAH) limits the conversion of phenylalanine to tyrosine. Without intervention, intellectual disability, significant delays in developmental milestones, hyperactive behaviour with autistic features, and seizures may occur. High levels of brain phenylalanine are probably the main cause of neurotoxicity [1] by interfering with cerebral protein synthesis [2], increasing myelin turnover and inhibiting neurotransmitter synthesis [3,4]. In children with classical PKU, their only effective management option is a severely restricted low protein/phenylalanine diet that aims to lower blood phenylalanine levels to within a strict target range [5]. Although alternative treatments, such as sapropterin and pegvaliase (PEGylated recombinant *Anabaena variabilis* phenylalanine ammonia lyase (PAL)), have been licensed, they are only suitable for subsections of the PKU population, and access is restricted in some countries. Thereby, outcome in PKU is dependent on the early introduction of dietary treatment and the quality of lifelong blood phenylalanine control, which in turn is determined by the ability to adhere to dietary treatment.

A low protein diet aims to prevent long-term phenylalanine toxicity, with most patients tolerating < 10 g/day of natural protein [6]. All high biological protein foods (e.g., meat, eggs, fish, ordinary bread, pasta and flour) are not allowed [7]. The diet is supplemented with a minimal/free phenylalanine synthetic protein (protein substitute), which in classical PKU provides approximately 80% of total protein intake [8]. In the UK, the protein substitute is given with an allocated daily amount of measured phenylalanine from a range of regular foods (e.g., potatoes, peas, cereals) to meet essential requirements. Low protein foods are given without restriction and include low protein regular foods (containing protein ≤ 0.5 g/100 g), fruit and vegetables (containing phenylalanine ≤ 75 mg/100 g), and special low protein foods (SLPFs) (containing phenylalanine ≤ 25 mg/100 g) [5,9]. 

In a low phenylalanine diet, SLPFs are essential safe foods, satisfying satiety, offering choice and replicating some normality in a lifelong restricted diet [7]. They are categorised as ‘Foods for Special Medical Purposes’ (FSMPs) and are defined as specialist foods for the dietary management of patients with a medical condition who are unable to achieve a suitable nutritional intake from regular foods. They are described as ‘evidence based nutritional solutions for disorder related conditions’ [10]. They are ‘highly regulated’ by European Union law, with ‘*Delegated Regulation (EU) No 2016/128*’ setting in place policies on composition and labelling. FSMPs should be used only under medical/health professional supervision and must be labelled according to their intended use. 

SLPFs replace basic food items such as milk, bread, and pasta. They help optimise growth, provide energy to prevent catabolism, and avoid consequential raised blood phenylalanine [11]. A wide range of SLPFs may be key in helping patients sustain their dietary treatment for life, although access to SLPFs varies across Europe (European Society for Phenylketonuria (ESPKU)) [12]. Pena et al. [13] showed that the availability of SLPFs in different countries in Europe ranges from 73 SLPFs in Portugal to 256 in Italy, while no information was available for some countries. Only a few specialist companies manufacture SLPFs, and due to the constraints of a phenylalanine restricted diet, their nutritional composition consists mainly of carbohydrate and fat. Their taste and aesthetic properties are prioritised over their nutritional composition, although producing acceptable, high-quality SLPFs from isolated food starches with good organoleptic properties, texture and colour is challenging. While the variety and availability of SLPFs has improved, the choice, ease of access and quality in comparison to regular foods are all still narrow, inflexible and inadequate. There is only limited data about the nutritional contribution made by SLPFs to a phenylalanine restricted diet, or the types of SLPFs that are consumed.

In a longitudinal 3-year prospective study, the aim was to evaluate the contribution of SLPFs to the energy and macronutrient intake of a group of well-controlled children with PKU on dietary treatment only. As a secondary aim, we examined the dietary patterns of children when using SLPFs routinely in their diets. 

## 2. Materials and Methods

In total, 50 children (28 boys, 22 girls) with PKU were recruited. Of these children, 47 were European and 3 were of Pakistani origin. Study inclusion criteria: diagnosed by newborn screening; aged 5–16 years; not treated with sapropterin dihydrochloride; and 70% of blood phenylalanine concentrations within target range for 6 months before study enrolment. The target blood phenylalanine ranges for children aged 5–12 years were 120 to ≤360 µmol/L, and for 12 years and older they were 120 to ≤600 µmol/L, as recommended by the European PKU guidelines [5]. Based on untreated blood phenylalanine levels at newborn screening (<1000 µmol/L) and dietary phenylalanine tolerance (>750 mg/day), the majority of participants had classical PKU, except two children with mild PKU.

### 2.1. Protein Substitute Intake

At enrolment, the protein substitute sources were casein glycomacropeptide (CGMP-AA), *n* = 31, and amino acid supplements (AA), *n* = 19. The AAs were either ready-to-drink liquid pouches providing 10, 15 or 20 g protein equivalent or powders made up with water to a semi-solid consistency. The CGMP-AA was a powder mixed with 120 mL water. It contained 20 g of protein equivalent, with a residual amount of phenylalanine (36 mg/20 g protein equivalent) and it was given as a drink. 

Dietary intake was assessed by 2 types of dietary assessment technique:Three-day recorded food diary: caregivers were instructed on how to record food intake using scales, household measures or from a pictorial handbook with measured food portion sizes. A three-day semi-quantitative dietary assessment was completed, with an annual mean of 4 (range 3–6) assessments per child for a period of 3 years. Assessments were checked via face-to-face interviews by one of two trained metabolic dietitians. Portion weights of SLPFs were provided by manufacturers’ information or estimated from the ‘low protein’ portion size picture book. At least once annually, children were observed eating one meal at home, with portion sizes weighed and checked.Food frequency questionnaire (FFQ): the FFQ, specifically designed for patients with PKU, contained a series of questions on the consumption of both SLPFs and regular foods, estimating the portion sizes eaten and frequency of consumption of each food item. Foods were grouped into dairy products, cereals, fats, sugar and sweet foods, drinks, fruit and vegetables, and ‘meat, fish, eggs’, with ‘free from’ or SLPF alternatives for each category. The FFQ diaries were completed at the same time as the 3-day diet diaries, with a mean of 3 (range 3–6) FFQs completed yearly, constructing a comprehensive database on the actual consumption of special low protein and regular foods.

All the dietary assessments were analysed using Nutritics Nutritional Software (v5.093) [14]. The results were compared with age and gender specific UK dietary reference values and estimated average requirements for energy (EAR) (UK Scientific Advisory Committee on Nutrition (SACN) [15]. The database included the nutrient analysis of protein substitutes and SLPFs, using nutritional information supplied by the manufacturers.

The contribution of SLPFs to macronutrient intake was calculated by age for children ≤11 years and those ≥12 years. This age range was chosen to match with the age-dependent upper target blood phenylalanine concentrations given by the European PKU Guidelines [5]. At the point that children reached ≥12 years of age, they were then transferred to the older age group. The following macronutrients were analysed: energy (Kcal), protein (g), carbohydrate (g), starch (g), sugar (g), fat (g) and fibre (g). For each subject for each year, the mean contributions of each macronutrient from protein substitute, SLPFs and regular foods were calculated, and the mean value for all subjects is presented. The annual mean total energy intake (Kcal/day) and % EAR has been compared with UK dietary reference values or EAR for energy [15]. From the FFQ, the regularity of meals (breakfast, midday and evening meal), frequency of snacks and drinks, and the amount of foods consumed each week were also estimated. The FFQ was used to calculate the number of portions in grams of each food from the different food groups. Using this data, the mean number of grams of food eaten each week was calculated. This data complemented the dietary assessment analysis, showing the foods that were consumed regularly, but also highlighting any foods that might have been omitted from the 3-day dietary assessments. This data was not used to estimate energy or nutrient intake and was not statistically analysed, but showed the typical weekly pattern of foods consumed, and how meals were structured.

### 2.2. Anthropometric Measurements

Weight, height and BMI were measured once every 3 months by one of two metabolic dietitians. Height was measured with a Harpenden stadiometer (Holtain Ltd., Crymych, UK) and weight on calibrated digital scales (Seca, Medical Measuring Systems and Scales, Birmingham, UK model 875); both were measured to the nearest 0.1 cm or kg, respectively.

### 2.3. Blood Phenylalanine Levels

Trained parents/caregivers collected weekly early morning fasted blood spots on filter cards, Perkin Elmer 226 (UK Standard NBS). Blood specimens were sent via first class post to the laboratory at Birmingham Children’s Hospital. All the cards had a standard thickness, and the blood phenylalanine concentrations were calculated on a 3.2 mm punch by MS/MS tandem mass spectrometry. At enrolment, the median blood phenylalanine concentrations for the previous 12 months were collected and referred to as the enrolment blood phenylalanine concentration.

### 2.4. Statistical Analysis

Mann Whitney nonparametric unpaired t tests comparing two unmatched groups of data were used to compare macronutrient differences (energy, protein, carbohydrate, starch, sugar, fat and fibre) between children ≤11 years and those ≥12 years. The quantitative outcome measures have been summarised and descriptive statistics reported as means and differences assessed between the groups, with a statistically significant value of *p* < 0.05.

### 2.5. Ethical Permission

The South Birmingham Research Ethics (REC) committee granted a favourable ethical opinion, referenced REC13/WM/0435 and IRAS (Integrated research application system) ID 129497. Written informed consent was obtained for all subjects from at least one caregiver with parental responsibility, and written assent obtained from the subject if appropriate for their age and level of understanding.

## 3. Results

In total, 48 children (21 girls and 27 boys) completed the study. The mean age at enrolment was 9.3 years (5–16 years). There were 35 children aged ≤11 years and 13 aged ≥ 12 years. 

### 3.1. Subject Withdrawal

One boy and one girl (aged 12 years) were excluded from the study as both failed to comply with the study protocol. One failed to return blood phenylalanine samples and both had poor dietary adherence.

### 3.2. Dietary Prescription

Over the study period the total mean daily dose of protein equivalent from protein substitute was 64 g/day (range 40–80 g) or 1.5 g/kg (1–2 g/kg) with the mean amount of prescribed natural protein 5.5 g protein/day (range 3–30 g) or 275 mg phenylalanine (range 150–1500 mg)/day. The protein substitute source was AA, *n* = 19 (liquid pouches (PKU Lophlex LQ, Nutricia Ltd. Trowbridge, UK. *n* = 1; PKU Cooler, Vitaflo International Ltd., Liverpool, UK. *n* = 14), or powder (PKU gel, Vitaflo International Ltd., *n* = 4)). In total 29 children took CGMP-AA (GMP study product, Vitaflo International Ltd.); *n* = 13 had their entire protein substitute requirement as CGMP-AA, and *n* = 16 took a combination of CGMP-AA and AA. The numbers of children taking AA products in combination with CGMP were liquid pouches *n* = 15, (PKU Lophlex, Nutricia, *n* = 4; PKU Cooler, Vitaflo International Ltd., *n* = 11), and for those taking powder (PKU gel, Vitaflo International Ltd.) *n* = 1.

### 3.3. Energy and Macronutrient Intake

Mean daily energy intake and % EAR are described in Table 1. For both age groups the energy as a percentage of EAR was age appropriate and within 5% of the EAR. In both groups the total percentage contribution of energy from carbohydrate, protein and fat was similar. 

### 3.4. Contribution of SLPFs to Mean Macronutrient Intake

#### 3.4.1. Energy Intake

The total mean energy intake in the combined age groups was 2059 Kcal/day, of which the percentage mean energy intake from SLPFs was 33% (SD ± 8), regular foods 42% (SD ± 13) and protein substitute 21% (SD ± 5) (Table 1). Of the total energy, 2% (SD ± 3) was provided by phenylalanine/natural protein containing foods (potatoes, crisps, and vegetables with a phenylalanine content ≥ 75 mg/100 g protein).

The contribution of SLPFs to the mean daily intake of carbohydrate was 40% (SD ± 10), starch 51% (SD ± 18), sugar 21% (SD ± 8) and fat 38% (SD ± 13). The daily intake of sweet SLPFs (e.g., biscuits, cakes, and chocolate) was low, and overall contributed minimally to the energy, fat or carbohydrate intake. Table 2 describes the mean contribution of energy and grams per day from carbohydrate (starch and sugar), fat and protein to total daily energy intake. 

#### 3.4.2. Carbohydrate Intake

The mean intake of carbohydrate from SLPFs, regular foods and protein substitute for all children was 294 g/day, of which starch provided 174 g/day (59%) and sugar 119 g/day (40%) (Table 2). 

SLPFs were the highest contributor to total carbohydrate intake, with a mean intake from bread of 50 g/day (17%), pasta of 39 g/day (13%) and low protein milk replacement of 9 g/day (3%). The highest contribution from regular foods to mean carbohydrate intake came from drinks (carbonated and cordials) at 20 g/day (7%), potatoes at 19 g/day (6%), fruit at 14 g/day (5%), and crisps and confectionary, both providing 12 g/day (4%). Protein substitute contributed 26 g/day (9%) to the total mean carbohydrate intake. Other foods making up the total carbohydrate are shown in Table 3. The only significant difference in the intake of SLPFs between children aged ≤11 and ≥12 years was for the low protein milk replacement, this being higher in the younger age group (*p* < 0.0001). 

The combined age groups had a mean daily starch intake of 174 g/day. The highest starch intakes were provided from low protein bread, 37 g/day (21%) and pasta/rice, 34 g/day (20%). Potato and crisps (phenylalanine containing foods) were the other main non-SLPF contributors to starch intake (Table 3). Protein substitutes provided 11 g/day (6%) of the total starch intake.

Sugar intake supplied by SLPFs was minimal. In the combined age groups, low protein bread provided a mean intake of 4 g/day (3%). Sugar from low protein milk provided 5 g/day (4%), with a higher intake in children ≤11 years of age. The regular foods contributing to sugar intake were sweet drinks, providing 17 g/day (14%), fruit, 14 g/day (12%), and confectionary, 10 g/day (8%). Protein substitutes provided 18 g/day or 15% of the total sugar intake. The total mean amount of free sugar, defined as sugars added to cooked or manufactured food, was 26 g/day (22%), and this came largely from sweet drinks, e.g., cola, lemonade, sweets, jams, honey and condiments such as tomato sauce. The daily amount of free sugar, particularly from sweet drinks, was higher in children aged ≥12 years. Free sugars represented 5% of total energy intake, in line with SACN recommendations [16]. Fruit provided a high sugar intake, but this was from natural rather than refined sugars.

#### 3.4.3. Fat Intake

Fat intake provided a mean of 63 g/day for all children, with SLPFs supplying minimal fat intake. In the combined ages, bread supplied a mean fat intake of 7 g/day (11%), and in children ≤11 years milk contributed 5 g/day (8%). The highest fat sources were from butter and oils, 13 g/day (21%), potato crisps, 5 g/day (8%), and fried potatoes, 3 g/day (5%). The main sources of fat were saturated fat from oil/butter, fried potatoes and crisps. Protein substitutes provided a mean fat intake of 4 g/day (6%), with some containing essential fatty acids and/or long-chain polyunsaturated fatty acids (LCPUFAs). 

#### 3.4.4. Protein Intake

SLPFs made no significant contribution to protein intake, while the protein equivalent from protein substitute consistently provided a mean intake of 64 g/day, which was 86% (SD ± 9) of the total protein intake. 

#### 3.4.5. Fibre Intake

SLPFs provided approximately 50% (SD ± 23) of the mean daily fibre intake. Low protein bread and pasta provided higher fibre sources than potatoes, vegetables and fruits. In children aged ≤11 y, the total mean fibre intake was 18 g/day, providing 83% of the recommended intake (SACN) [16], with 9 g/day (50%) from SLPFs. Children aged ≥12 years consumed a mean fibre intake of 20 g/day, providing 82% of the recommended intake, of which 11 g/day (55%) came from SLPFs. The blend of fibre was limited to the fibre sources added to SLPFs, which was commonly derived from gums and hydrocolloids. 

### 3.5. Median Blood Phenylalanine Concentrations throughout the 3-Year Study Period

Statistically, the phenylalanine concentrations were significantly different both within and between the groups from enrolment to year 3 (Table 4). This group of children were well controlled, with median blood phenylalanine levels within the European PKU guidelines [5]. There was no correlation between energy intake from SLPFs and phenylalanine concentrations.

## 4. Anthropometry

At the 3-year follow up, the median weight, height and BMI Z scores (range) were 1.0 (0.3–1.7), 0.3 (−0.01–0.6) and 1.1 (0.5–0.8) respectively (Table 5). Using the WHO [17] definitions of overweight and obesity, between enrolment and year 3, overweight (defined as BMI one standard deviation over the reference median) increased from 25% (*n* = 12/48) to 29% (*n* = 14/48), and obesity (BMI equivalent to two standard deviations over the reference median) increased from 10% (*n* = 5/48) to 17% (*n* = 8/48).

### Food Patterns from the Food Frequency Questionnaires

Children aged ≤11 years ate regular main meals, including breakfast, midday and evening meal, with a mean of two snacks per day. Children aged ≥12 years were more independent, and some would cook their own meals, usually based on pasta or bread. They had irregular meal patterns with less supervision around mealtimes. They commonly missed breakfast, eating snack foods for their midday meal particularly when in school and eating more food towards the evening after the school day had finished.

Over the 3-year study period, low protein milk replacement decreased in children aged ≤11 years from a mean of 1750 mL/week to 1300 mL/week, remaining consistent at 700 mL/week in children aged ≥12 years. In the younger age group, the mean low protein bread intake increased from 670 g/week to 750 g/week at year 3, based on an average bread slice weighing 30 g. This was equivalent to three to four slices/day. Low protein bread intake remained consistent in children aged ≥12 years of age, at 900 g/week (four to five slices/day). The mean intake of low protein pasta in children aged ≤11 years was 600 g/week based on an estimated cooked portion of 200 g (three portions/week). This increased in children aged ≥12 years from 700 to 900 g/week (four portions/week).

The overall daily fruit intake was low. In children aged ≤11 years, the mean intake was 1200 g/week (two portions/day), and 700 g/week (one portion/day) in children ≥12 years. Vegetable intake decreased over the 3 years in both groups, decreasing in children aged ≤11 years from a mean of 660 g/week to 560 g/week (approximately one portion/day) and from 900 g/week to 700 g/week (one to two portions/day) in children ≥ 12 years. Children aged ≤ 11 years consumed a mean intake of 1200 mL/week of sweet drinks (mainly from fizzy drinks), with an average serving of 200 mL (one sweet drink/day), whilst the older group drank 2300 mL/week (two sweet drinks/day).

The SLPFs that were regularly eaten by the entire group of children were bread, at 92% (*n* = 44/48), pasta, at 85% (*n* = 41/48), and low protein milk at 77% (*n* = 37/48).

## 5. Discussion

This study demonstrated that SPLFs were an essential energy source, providing over 30% of energy intake in children with PKU aged 5 to 16 years of age. The low protein staple foods bread and pasta made the largest consistent contribution to energy intake. There were few differences in SLPF intake between children ≤ 11 and ≥ 12 years of age, the exception being the younger children who consumed more low protein milk replacement. Protein substitutes provided 18 g/day (15%) of the total sugar intake. Concern has been expressed about the energy content of SLPF snack foods [13], but in this study, low protein cakes, biscuits and chocolate made a minimal contribution to daily energy intake. Instead, aspartame-free sweet drinks provided the highest intake of free sugars. Some may argue that the sugar content of protein substitutes is too high; however, in children it is important to provide a source of energy to ensure nitrogen is used efficiently.

Very few studies have examined the energy contribution made by SLPFs. An Italian study, in children with PKU aged 5 to 11 years, reported that SLPFs provided 47% of energy intake [18]. In a small German study, reporting on eight children aged 6 to 16 years of age, when on dietary treatment only, the SLPFs provided 39% of the energy intake [19], a higher energy intake than was observed in our study. Throughout Europe, SLPFs are available through a number of systems, including state national health schemes (either prescription or monthly financial family allowance), or in some countries patients/carers may be expected to make a complete or partial contribution to their purchase. It is unknown how these systems or patient acceptance impact the usage of SLPFs. In addition, it is also unknown how adherence to a phenylalanine-restricted diet and the overall quality of blood phenylalanine alters the usage of SLPFs.

In the UK, access to SPLFs is controlled. The National Society for PKU (NSPKU) provides age-defined guidance on the maximum monthly units of SLPFs that can be prescribed by community general practitioners (GPs). This is based on the assertion that SLPFs provide up to 50% of energy intake. It has been reported that both the NHS authority (e.g., the Clinical Commissioning Group or Health Board) and GPs have refused to prescribe or have limited the amounts of SLPFs that patients can access [6,20]. In some cases, patient requests for low protein cake mixes, or low protein cereals bars, have been rejected, even though our study indicates that these contribute a negligible energy intake. Both Cochrane et al. and Ford et al. [6,20,21] have described the stigma caregivers and patients encounter when obtaining SLPFs via their GP. Due to access issues, on occasion patients are without these foods, leading to anxiety about food insecurity, which has recently been reported in PKU [22]. Even in the early 1950s, when dietary treatment commenced, it was recognised that catabolism led to increased blood phenylalanine concentrations, and therefore an adequate energy intake supplied by SLPF is a necessity [23].

It has been suggested that the uncontrolled consumption of SLPFs may cause obesity [13,18], although the principle low protein foods eaten by our patient cohort were bread and pasta. However, as in other studies [10,24], our children consumed a low fat, high carbohydrate diet, leading to an imbalance in macronutrient composition. Despite energy intake only meeting recommendations, both overweight and obesity increased over the 3-year study. It has been observed from the age of one year that the energy provided by carbohydrate is higher than in healthy controls [24]. Clearly, a balanced diet prevents co-morbidities, such as metabolic syndrome, obesity, coronary heart disease and diabetes type II [17,25,26]. The type of carbohydrate is also an important health consideration. Insulin resistance measured by HOMA-IR (Homeostasis Model Assessment Insulin Resistance) has been shown to be higher in subjects with PKU [27], especially those who are overweight [27] or those with central obesity [28]. Similarly, the dietary glycaemic index and load was higher in children with PKU, suggesting a link between the quality of carbohydrate and peripheral insulin resistance [18]. Furthermore, lower total/LDL and higher triglyceride/HDL cholesterol ratios have been reported in children with PKU, suggesting an association between the quality of carbohydrate and triglyceride glucose index [28].

The starch sources from SLPFs, bread and pasta, eaten by children in our study were derived from starch isolated from wheat, maize and rice. Isolated starches are refined, having different physiologic properties compared to complex starch forms, and foods containing these may have a higher glycaemic index than those made from wheat flour [29,30]. However, a high intake of sugar from regular sweet drinks is also problematic. Many ‘sugar free’ drinks are unsuitable for children with PKU as they contain aspartame, a source of phenylalanine, limiting choice and increasing the glycaemic index of foods consumed. Importantly, the aspartame content of drinks may vary significantly, and the phenylalanine content is not identified on the food label [31].

Fibre sources may alter the gut microbiome, increasing the risk of chronic diseases such as inflammatory bowel disease and obesity [32]. The main fibre sources added to low protein bread and pasta were hydrocolloids. These are common additives in the food manufacturing industry, aiding texture and viscosity, but their role in gut health is limited [33,34]. Although the health benefits of hydrocolloids have been reported, there is little understanding of how these function in the intestine, or of their physiological benefits [35]. The fruit, vegetable and fibre intake of children in this study was less than the UK government’s ‘5 a day’ healthy eating recommendations, derived from World Health Organisation (WHO) and SACN recommendations [16,17,36]. Fruits and vegetables low in phenylalanine (≤75 mg/100 g), except potatoes, make a valuable contribution to the dietary intake, as they can be eaten ad libitum [37]. The free consumption of these fruits and vegetables does not impact metabolic control and should be encouraged in a low phenylalanine diet as a source of beneficial fibre. Cereal and wholegrain fibres, associated with a lower risk of cardio metabolic disease and colorectal cancer, and promoted by SACN [16], are precluded in a phenylalanine restricted diet. The challenge and responsibility of manufacturers making SLPFs is to provide a source of beneficial fibre maximising gut microbiome health.

There are limitations to this study. Firstly, all dietary assessment methods are open to misinterpretation. To minimise error, the standard weights of foods were collected regularly, and at least one meal was observed and food items weighed by the same dietitian. The food frequency questionnaire was not validated, although two dietary assessment tools were used, and mealtime portion sizes were observed and weighed by a dietetic researcher to help improve the quality of the dietary data collected. The nutritional composition, including the starch and sugar content, of SLPFs was not always available on food labels, and there were some discrepancies between food labels and manufacturers’ websites [29]. It was not possible to accurately assess the salt intake from food labels or from the amounts added to food in cooking or at the table.

## 6. Conclusions

In children with PKU, dietary intake is based on a lower number of regular foods, offering limited variety. This study showed that SLPFs make an important contribution to energy intake in a phenylalanine restricted diet, with consistent dietary patterns over time demonstrating long-term dependence on essential foods, such as low protein bread, pasta and milk. The intake of sugar and fat from SLPFs was minimal. SLPFs should be unlimited to all patients on a phenylalanine-restricted diet, helping their ability to sustain their dietary restriction and reducing anxiety around food insecurity. Further improvements in the nutritional quality of the diet would aid in securing longer-term health benefits and adherence to a severe lifelong regimen.

## Figures and Tables

**Table 1 nutrients-12-03153-t001:** Mean (standard deviation) total energy intake, percentage of EAR, and the contribution of carbohydrate, protein and fat as a percentage of mean total energy intakes in children ≤ 11 years and ≥ 12 years, and for the combined age groups over the 3-year study period.

Year 1 to 3 Mean Intake (SD)	≤11 years *n* = 35	≥12 years *n* = 13	All Children (≤11 ≥12 years) *n* = 48
Energy intake Kcal/day	1921 (255)	2224 (417)	2059 (394)
% EAR	105 (21)	95 (13)	99 (15)
**Mean % (±SD) energy contribution from carbohydrate, protein and fat**			
Carbohydrate	58 (4)	56 (7)	57 (5)
Protein	15 (3)	14 (4)	14 (4)
Fat	27 (4)	28 (7)	28 (5)

EAR, estimated average requirement. SD, standard deviation. EAR for energy was 2175 kcal (1422–2809) [15]. Calculated by taking the median (range) for the combined ages and gender. Kcal Kilocalories.

**Table 2 nutrients-12-03153-t002:** Mean (standard deviation) contribution of calories and grams per day from carbohydrate (including starch and sugar), fat and protein to mean total energy intake for all children over the 3-year study period.

	All Children (≤11 ≥12 years) *n* = 48	
Macronutrient	Mean kcals/day (±SD)	Mean g/day (±SD)
Energy	2059 (394)	-
CHO	1176 (60)	294 (15)
Starch	687 (24)	174 (5)
Sugar	477 (20)	119 (5)
Fat	576 (36)	63 (4)
Protein	307 (5)	74 (2)

SD, standard deviation. CHO, carbohydrate.

**Table 3 nutrients-12-03153-t003:** Mean contribution of SLPFs, regular foods and protein substitute in grams to total carbohydrate, starch, sugar and protein intake for children with PKU.

Macronutrient	CHO (g)		Starch (g)		Sugar (g)		Fat (g)		Protein (g)		CHO (g)	Starch (g)	Sugar (g)	Fat (g)	Protein (g)
Age	≤11 years	≥12 years	≤11 years	≥12 years	≤11 years	≥12 years	≤11 years	≥12 years	≤11 years	≥12 years	All Ages				
Total macronutrient intake from all foods and PS	274 g	313 g	165 g	182 g	110 g	128 g	57 g	68 g	71 g	77 g	294 g	174 g	119 g	63 g	74 g
SLPFs															
Bread	43	56	34	40	3	4	6	7	-	-	50	37	4	7	-
Pasta/Rice	38	40	30	38	-	-	1	1	-	-	39	34	-	1	-
Milk substitute	14 ^§^	3 ^§^	2	1	7	3	7	2	-	-	9	3	5	5	-
Biscuits	5	4	3	2	2	1	2	2	-	-	5	3	2	2	-
Cereal bars	5	4	3	2	8	6	3	1	-	-	3	4	7	2	-
Cakes/puddings	6	4	4	2	4	3	5	3	-	-	5	5	3	4	-
Miscellaneous: burger, sausage, pizza, homemade dishes	11	14	3	5	3	5	2	4	-	-	6	3	4	3	
Total macronutrients from SLPFs	122 g	125 g	79 g	90 g	27 g	22 g	26 g	20 g	-	-	117 g	89 g	25 g	24 g	-
Regular foods															
Potato	16	22	17	20	2	4	3	4	2	3	19	19	3	3	2
Crisps	11	13	12	13	-	-	4	5	1	2	12	12	-	5	2
Cereals	9	6	7	4	1	1	-	-	1	1	7	5	1	-	1
Dairy	3	3	1	1	1	1	1	1	-	-	3	1	1	1	-
Vegetables	3	6	-	-	5	5	2	4	1	2	5	-	5	3	2
Fruit	14	13	-	-	15	13	-	-	1	1	14	-	14	-	1
Sweets/chocolate	8	15	2	5	5	12	2	4	-	1	12	4	10	3	-
Cereal bars	3	3	2	2	2	3	1	1	-	1	3	2	3	1	1
Cakes/puddings	5	3	3	3	3	3	1	1	-	-	4	3	3	1	-
Drinks	14	25	-	-	14	20	-	-	-	-	20	-	17	-	-
Butter/oils	-	-	-	-	-	-	11	14	-	-	-	-	-	13	-
Sauces, jam	24	31	24	29	13	19	2	4	-	1	31	17	16	4	1
Total macronutrients from regular foods	110 g	140 g	68 g	77 g	61 g	81 g	27 g	38 g	6 g	12 g	130 g	63 g	73 g	34 g	10 g
Protein substitute	27 g	25 g	13 g	9 g	18 g	17 g	3 g	4 g	65 g	64 g	26 g	11 g	18 g	4 g	64 g
Total intake from all food and PS	259 g	290 g	160 g	176 g	106 g	120 g	56 g	62 g	71 g	76 g	273 g	163 g	116 g	62 g	74 g
Total % intake	95%	93%	97%	97%	96%	94%	98%	91%	100%	99%	93%	94%	97%	98%	100%

§ *p* ≤ 0.0001, CHO carbohydrate, PS protein substitute, SLPFs special low protein foods. NB figures do not add to 100% due to inaccurate measurement of CHO, starch and sugar from SLPF information.

**Table 4 nutrients-12-03153-t004:** Median (range) phenylalanine concentrations in children aged ≤ 11 years and ≥ 12 years at enrolment and follow up at 3 years.

Median Phenylalanine µmol/L	≤ 11 years (*n* = 35)	≥ 12 years (*n* = 13)	*p* Value
Enrolment (range)	270 µmol/L *^§^ (140–470)	356 µmol/L * (230–600)	* *p* = 0.003
3 year follow up (range)	300 µmol/L **^§^ (200–730)	485 µmol/L ** (320–895)	** *p* < 0.0001 ^§^ *p* = 0.02

*, **, ^§^—*p*-values between and within the groups.

**Table 5 nutrients-12-03153-t005:** Median (range) annual weight, height and BMI Z scores for all children at enrolment and follow up at year 3.

Follow up Duration year	Weight Z Score	Height Z Score	BMI Z Score
Enrolment	0.7 (−0.1–1.2)	0 (−0.2–0.5)	0.7 (0.0–1.2)
year 1	0.8 (0.3–1.4)	0.2 (−0.3–0.4	1.0 (0.3–1.5)
year 2	0.9 (0.4–1.7)	0.2 (−0.1–0.6)	1.0 (0.3–1.9)
year 3	1.0 (0.3–1.7)	0.3 (−0.01–0.6)	1.1 (0.5–1.8)

BMI: Body mass index.

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
