# Peer review of "A 3 Year Longitudinal Prospective Review Examining the Dietary Profile and Contribution Made by Special Low Protein Foods to Energy and Macronutrient Intake in Children with Phenylketonuria"

_nutrients, 2020, doi:10.3390/nu12103153_

Round 1

Reviewer 1 Report

In general: good written, easy to read manuscript. Relevant, as the findings are important for  the discussion on acces, availability and financing of these special low protein foods. In addition it adds to the importance regulation of these foods

Introduction

strong: The background of the special low portein foods in PKU is nicely described and invites the reader to read the rest of the manuscript. 

suggestion: Maybe the introduction can be sligtly shortened, some parts on availability might be added to he discussion instead.

Methods

Strong: The authors were able to include a relatively large number of patients with a lot of data on diet.

some minor points:

where there patients starting in group <11 on enrollment but later in group >12 (for example a 10 year old on enrollment.  would be in group >12 at the end of the study)?

How frequent were the Phe spots done? What were the antropometrics of this group?

Maybe table 2 and 3 can be combined

Table 5: I miss a footnote on the meaning of **

Discussion: The discussion nicely decribes what the data of this study brings us.

Minor point: this is a very well controlled population and for less well controled populations the data might be different

layout: Line 304: the sentence is interrupted en continues on the next line

Author Response

10th October 2020

Dear Editor

Thank you for your comments. We have made the changes to the manuscript as suggested.

Comment 1: strong: The background of the special low portein foods in PKU is nicely described and invites the reader to read the rest of the manuscript.

Suggestion: Maybe the introduction can be sligtly shortened, some parts on availability might be added to he discussion instead.  

Thank you we have shortened the introduction and added most of the section on availability to the discussion.

Comment 2: where there patients starting in group <11 on enrollment but later in group >12 (for example a 10 year old on enrollment.  would be in group >12 at the end of the study)?  

We have explained this more clearly in the methodology, but from the time children reached their 12 th birthday, their data onwards was included in the age group ≥12 years.

Comment 3: How frequent were the Phe spots done?

We have added in the frequency of blood phenylalanine monitoring into methodology section.

Comment 4: What were the antropometrics of this group?

Thank you. We have included the anthropometric details for the whole group, and added this to a table.

Comment 5: Maybe table 2 and 3 can be combined

Thank you we have considered this but table 3 is for all the children, and table 2 for the groups of children by age. However, we have combined table 1 and 2 making it easier for the reader.

Comment 6: Table 5: I miss a footnote on the meaning of **

Thank you we have added these to the table 

Comment 7: The discussion nicely decribes what the data of this study brings us.

Minor point: this is a very well controlled population and for less well controled populations the data might be different.

Thank you. We agree with the referee and added a comment about this in the discussion.

Comment 7: layout: Line 304: the sentence is interrupted en continues on the next line

Thank you this has been corrected

Reviewer2

Comment 1; There should be a space between the numbers and the units

Thank you we have corrected this through out the text.

Comment 2: Removal of Table from the paragraph

Thank you we have removed the word ‘table’ from the paragraph headings.

Comment 3: EAR to be provided

Thank you we have provided the median intake for the combined age and genders with a range and added this to table 1.

 Comment 4: References are not formatted correctly

 Thank you. We have reformatted the references.

Reviewer 2 Report

The study was properly designed and well described. This work provides an interesting contribution to the current state of knowledge.

I have only minor comments that should be taken into account during the revision:

There should be a space between the numbers and the units (e.g. grams). Please make corrections throughout the text.

Line 177: please remove "(Table 1, 2)" from the name of the paragraph.

Line 177-180: the reference for EAR should be provided.

References are not formatted correctly, please correct.

Author Response

(The authors gave the same response as above.)
